



# Biomass burning CCNs enhance the dynamics of a Mesoscale Convective System over the La Plata Basin: a numerical approach

Gláuber Camponogara[1], Maria Assunção Faus da Silva Dias[1], and Gustavo G. Carrió[2]

[1]Instituto de Astronomia, Geofísica e Ciências Atmosféricas da Universidade de São Paulo, Brazil
[2]Department of Atmospheric Sciences of Colorado State University, USA

*Correspondence to:* G. Camponogara (glauberic@gmail.com)

**Abstract.** High aerosol loadings are discharged into the atmosphere every year by biomass burning in the Amazon and Central Brazil during the dry season (July–December). These particles, suspended in the atmosphere, can be carried via a low level jet toward the La Plata Basin, one of the largest hydrographic basins in the world. Once they reach this region, the aerosols can affect mesoscale convective systems (MCS), whose frequency is higher during the spring and summer over the basin. The

5 present study is one of the first that seeks to understand the microphysical effects of biomass burning aerosols from the Amazon Basin on mesoscale convective systems over the La Plata Basin. We performed numerical simulations initialized with idealized CCN profiles for an MCS case observed over the La Plata Basin on 21 September 2010. The experiments reveal an important link between CCN number concentration and MCS dynamics, where stronger downdrafts were observed under higher amounts of aerosols, generating more updraft cells in response. Moreover, the simulations show higher amounts of precipitation as

the CCN concentration increases. Despite the model's uncertainties and limitations, these results represent an important step toward the understanding of possible impacts on the Amazon biomass burning aerosols over neighboring regions such as the La Plata Basin.

## 1 Introduction

The Amazon Basin exhibits a huge contrast in terms of aerosol concentration between wet (January-June) and dry (July-

15 December) seasons (Andreae et al., 2004; Martin et al., 2010; Artaxo et al., 2013). Aerosol loading is typically about hundreds of particles per cm$^3$ in the wet season with a similar pattern and microphysical characteristics to those found in remote ocean regions. For this reason, the Amazon region has been referred to as "green ocean" (Williams et al., 2002). However, during the dry season, when the Amazon Basin faces a polluted regime with tons of particles being released into the atmosphere by biomass burning, the number of aerosols increases drastically, close to one order of magnitude. The majority of these

20 emissions are due to anthropogenic activities regarding agriculture and deforestation (Reinhardt et al., 2001; Morton et al., 2008). According to Freitas et al. (2005), these aerosols can be transported by the wind to other regions such as the La Plata Basin.

The La Plata Basin is one of most intense convective regions in the world (Zipser et al., 2006) with significant lightning events (Albrecht et al., 2016). This region is dominated by mesoscale convective systems (MCSs) during the spring and summer





seasons (Velasco and Fritsch, 1987; Conforte, 1997; Torres and Nicolini, 2002; Vera et al., 2006; Salio et al., 2007; Durkee and Mote, 2010), which have an important role in the La Plata Basin's hydrological cycle, being responsible for approximately 90% of the total rainfall over the basin (Nesbitt et al., 2006). These systems may produce strong winds, floods, heavy rain and hail (Velasco and Fritsch, 1987; Fritsch and Forbes, 2001).

Mesoscale convective systems normally result from the combination of certain ingredients such as atmospheric instability, mesoscale circulation, the weak midlevel short-wave trough and low level jet (LLJ). Mesoscale circulations contribute to air parcels reaching the free convection level; atmospheric instability favors convection development; the midlevel short-wave trough intensifies low level convergence; and the LLJ brings moisture and heat from the Amazon (Fritsch and Forbes, 2001; Silva Dias et al., 2009). Moreover, the LLJ may also advect huge amounts of aerosols from the Amazon to the La Plata Basin

during the dry and dry to wet transition seasons (Freitas et al., 2005), which may interact with MCS in different ways.

Aerosols can absorb and scatter solar radiation (direct effect) leading to a decrease in the surface temperature (semi-direct effect) or even a stabilization of the atmosphere by warming the surrounding air (Eck et al., 1998; Koren et al., 2004, 2008). Part of the aerosols can have affinity with water and act as cloud condensation nuclei (CCN), this being known as the indirect or microphysical effect. In addition, CCNs can increase the warm cloud albedo for a constant liquid water content (Twomey,

1974) and change the lifecycle and the warm rain triggering mechanism (Albrecht, 1989; Martins et al., 2009).

As mentioned in the literature (Rosenfeld, 1999; Rosenfeld et al., 2008; van den Heever et al., 2006; Carrió and Cotton, 2011; Carrió et al., 2014; Zhou et al., 2016), the aerosol microphysical effect is related to the number of cloud droplets that are nucleated. For example, a polluted atmosphere (large number of CCN particles) produces more cloud droplets than a clean one if there is enough water vapor to support a larger population. This effect leads to a narrower cloud droplet size spectrum filled

with smaller cloud droplets, which delays the collection growth onset. Hence, more cloud droplets are thrust into freezing levels, becoming supercooled. Finally, the supercooled droplets may either be collected by ice particles (riming process) or freeze homogeneously producing graupel and eventually hail, and ultimately affecting rainfall at the surface.

By using the Brazilian development on the Regional Atmospheric Modeling System (BRAMS), Martins et al. (2009) investigated the aerosol effects on cloud microphysical processes during the end of the Amazon dry season and observed significant

impacts on precipitation process in spatial and temporal dimensions. The maximum liquid water values increased for high aerosol concentrations in all runs. By using a combination of different observational data sets, Gonçalves et al. (2015) noted that a high concentration of aerosol in the Amazon Basin may increase the cloud lifetime during the dry season as well as convection strengthening.

The invigoration of convective cells due to a CCN increase is usually explained by the latent heat release increase caused by

30 riming enhancement (van den Heever et al., 2006; Rosenfeld et al., 2008). However, CCNs also can affect cold pool and wind shear interaction, modulating the convection strength (Fan et al., 2009). When cold pool and wind shear strengths are balanced, convective cells tend to become more upright, and, consequently, more intense (Rotunno et al., 1988). Environments with high CCN concentrations may invigorate the convection by weakening the cold pool under weak shear conditions (Fan et al., 2009; Lebo and Morrison, 2014). Polluted atmospheres favor the development of fewer but larger raindrops, hence, evaporation

cooling is reduced, which decreases the cold pool strength. On the other hand, a combination of strong wind shear with high





aerosol loadings may overcome the latent heat release and, as a consequence, weaken convection. This occurs because stronger cold pools are needed to keep the updraft cells more upright under strong wind shear conditions, as shown by Rotunno et al. (1988).

Lebo and Morrison (2014) also observed different responses in precipitation when wind shear is intensified under polluted conditions. As wind shear increases, aerosols tend to generate more precipitation due to greater condensation. In contrast, precipitation was significantly reduced in the strongest wind shear scenario. This effect was caused by an excessively tilting downshear of updraft cells that led to raindrops to fall ahead of the gust front and thus to immediately evaporate.

Through a bidimensional cloud resolving model, Tao et al. (2007) examined the aerosol effects in three distinct deep convection cases from regions with different environments: sea breeze convection in Florida, USA; the squall line in Kansas, USA; and the tropical mesoscale convective system over the Pacific Ocean. For all cases, rainfall suppression was observed under high CCN concentrations during the initial stages of the systems. Conversely, in the mature stage, rainfall underwent suppression only over Kansas, being little affected over Florida and intensified over the Pacific. These results suggest that evaporation cooling plays an important role in this phenomenon since stronger (weaker) evaporation was observed in polluted environments under moister (drier) conditions. Stronger evaporation favors more intense downdrafts and cold pools, which, depending on atmospheric shear, may intensify the convection and then increase the precipitation.

According to Fan et al. (2007), large aerosol loadings are capable of greatly changing the convection strengthening and rain rate in environments with a high moisture content. On the other hand, Carrió et al. (2014) observed a stronger effect of CCN on hail under low level drier conditions, which favor higher cloud bases, and which do not contribute to warm rain processes, allowing more cloud droplets to become supercooled. As a result, the riming process is enhanced, and hail mass is increased. However, a further CCN increase may generate much smaller droplets, causing riming inhibition and, therefore, homogeneous freezing enhancement.

In summary, MCSs can be affected by aerosol and environmental conditions. Aerosol effects on MCSs can also be influenced by the environment, making the study of this phenomenon even more complex. As mentioned by Wall (2013), Camponogara et al. (2014) and Gonçalves et al. (2015), separating aerosol effects from environmental forcing is a great challenge. Therefore, despite the limitations, numerical models appear to be an important tool to understand the aerosol-cloud-precipitation processes, as pointed out by Tao et al. (2012). The present study is one of the first that seeks to understand the microphysical effects of biomass burning aerosols from the Amazon Basin on mesoscale convective systems over the La Plata Basin. In order to do so, we performed numerical simulations varying CCN number concentration for an MCS case over the La Plata Basin during the spring season. Section 2.1 gives a brief description of the atmospheric model, focusing on its microphysical parametrization, and Sect. 2.2 describes how the numerical experimentation was performed. Finally, results and discussion are presented in Sect. 3, followed by conclusions in Sect. 4.



## 2 Methodology

### 2.1 The Atmospheric Model

Originally developed from RAMS (Regional Atmospheric Modeling System), the Brazilian development on the Regional Atmospheric Modeling System (BRAMS), version 4.3, is used in this study. RAMS was created by a research group from
5 Colorado State University, joining three numerical models as reported by Cotton et al. (2003): a cloud/mesoscale model (Tripoli and Cotton , 1982); a hydrostatic version of this cloud model (Tremback, 1990); and a sea breeze model (Mahrer and Pielke, 1977). According to Cotton et al. (2003), RAMS, as well as BRAMS, is a non-hydrostatic model with several options related to its physics, which can be set according to the experiment type. It is possible to set different grid spacing using multiple grids, which can be nested either 1-way (coarser grids communicating with finer grids) or 2-way (both coarser and finer grids
communicating with each other). Detailed description of the current BRAMS version may be seen in Freitas et al. (2017).

BRAMS has been updated with the two moment microphysical bulk scheme currently used in RAMS, version 6.0, which predicts number concentration and mixing ratio for eight hydrometer species: cloud, drizzle, rain, pristine ice, snow, aggregates, graupel and hail. All hydrometeor categories have their own size distribution represented by a generalized gamma function (Walko et al., 1995; Meyers et al., 1997; Cotton et al., 2003; Saleeby and Cotton, 2004, 2008).

Together, cloud and drizzle categories represent a bimodal distribution of cloud droplets, as reported by Hobbs et al. (1980), and often observed in nature, where drizzle is basically large cloud droplets. The larger droplets function as an intermediate state between cloud droplets and raindrops, slowing the time that the cloud droplet takes to grow into raindrop size in a more realistic way. Pristine ices are primary ice crystals and grow only by vapor deposition. Once these ice crystals reach sizes greater than 100 $\mu$m they become snow. Aggregates are produced by collision and coalescence of pristine ice and snow species.
Only snow and aggregate categories can be converted to graupel. The high-density ice particles such as frozen raindrops and hailstone are represented by the hail category.

Cloud number concentrations are predicted by consulting a look-up table that was pre-computed through a bin-parcel model. This variable is a function of the air temperature, vertical velocity and the CCN number of particles. Drizzle number concentration can be produced by cloud droplet self-collection or Giant CCN activation (such as CCN activation). Pristine ice number
concentration is predicted by either IN activation, which depends on ice supersaturation, or homogeneous freezing. The aerosol concentrations can be defined as either a homogeneous (single vertical profile) or a heterogeneous field and can be advected and diffused.

The riming processes are computed by using a binned approach, which tends to be more accurate owing to considering an individual collection efficiency for each bin instead of taking just one collection efficiency for the entire gamma distribution
(Saleeby and Cotton, 2008). Bin sedimentation, sea salt and dust treatment and an algorithm for heat and vapor diffusion, without requiring iterations, are other examples of implementations currently present in the microphysical scheme (Cotton et al., 2003).

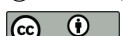



## 2.2 Experiments design

Numerical experiments were performed with BRAMS, version 4.3, for an MCS case observed over the La Plata Basin on 21 September 2010 (Fig. 3). We use three 1-way nesting grids with 16 km, 4 km and 2.5 km of grid spacing, respectively from the course to finest one. All grids are centered over the La Plata Basin, as can be seen in Fig. 1, and their domain sizes are

4800×5120 km (grid 1), 1400×1416 km (grid 2) and 1250×1250 km (grid 3). Figure 1 also shows a shaded area related to the La Plata Basin topography elevation. The vertical grid spacing varies from 100 m to 600 m, with a ratio of 1.1, whereas the top of domain extends to near 20 km, with 45 vertical levels. All experiments started on 20 September 2010 at 12:00 UTC. Table 1 shows the other important settings used in the experiments.

We performed four experiments with different vertical profiles of CCN concentrations, based on the work of Freitas et al.

(2005) (Fig. 2). The CCN profiles have their peaks at 2.5 km height, where CN-Low, CN-Mid, CN-High and CN-ExtHigh have 500, 1200, 1800 and 2800 particles per centimeter, respectively. Above 4.9 km, the CCN concentrations were considered $100 \, cm^{-3}$. These profiles were also nudged on the northern boundary of the finest grid domain, covering five meridional grid cells in order to maintain a constant aerosol input into the grid, as observed in northerly low level flow events (Freitas et al., 2005).

All the experiments were initialized heterogeneously by using the third generation of reanalysis from the National Centers for Environmental Prediction (NCEP) Climate Forecast System Reanalysis (CFSR). These data sets have 37 vertical levels, 4 times a day (00, 06, 12, 18 UTC), horizontal grid spacing of $0.5° \times 0.5°$ and are available for the period from 01/01/1979 up to 01/01/2011. Moreover, the model also was fed with the Normalized Difference Vegetation Index (NDVI), heterogeneous soil moisture, sea surface temperature weekly averaged, topography, land use and soil texture, downloaded from the *Centro*

*de Previsão do Tempo e Estudos Climáticos* of *Instituto Nacional de Pesquisas Espaciais* (CPTEC/INPE) through the link http://brams.cptec.inpe.br.

## 3 Results and discussion

Numerical experiments varying CCN concentrations (Fig. 2) were performed by using the BRAMS-4.3 model for the MCS case observed over the La Plata Basin on 21 September 2010. This mesoscale system was observed by the satellite GOES 12,

whose images can be visualized in Fig. 3. The first cells started over northeast of Argentina near 06:00 UTC. In the next few hours these convective cells begin to grow and become organized, forming a large cloud shield at 08:30 UTC. Near 11:00 UTC, the MCS reaches its mature stage, the South of Brazil almost entirely.

Synoptic fields are made from CFSR (Fig. 4) and used as initial and boundary conditions in BRAMS. Sea level pressure (contour lines) and thickness (shaded) are given by Fig. 4a. The thickness refers to the difference between the geopotencial

heights of the pressure levels of 500 hPa and 1000 hPa. This variable is proportional to the mean temperature in the layer between 500–1000 hPa. A trough, aligned meridionally, can be seen over the Argentina coast with a strong temperature gradient ahead (Fig. 4a). Furthermore, it is possible to note a warm air mass over the La Plata Basin, where a peak of thickness is found over northwest of Paraguay, the north of Argentina and the south of Bolivia, which is followed by a low pressure displaced





southward (Fig. 4a). Figure 4b shows the wind field (vectors) and specific humidity (shaded) at 850 hPa. A meridional flow is seen advecting moisture (Fig. 4b) and heat (Fig. 4a) from the Amazon Basin to Paraguay, northern Argentina and southern Brazil. The trough observed in Fig. 4a also appears in the wind field at 850 hPa as well as the South Atlantic High. Geopotential height (contour) and vertical p–velocity (shaded) at 500 hPa are given by Fig. 4c. The trough observed in at 1000 hPa and

850 hPa levels, also appears at 500 hPa, but slightly displaced to the northeast (Fig. 4c), followed by strong upward motions ahead with peaks of -2.5 Pa/s. A large region of significant negative values of p–velocity associated to the trough is observed over the Atlantic Ocean (50 °W – 35 °W and 50 °S – 33 °S) and the La Plata Basin. Figure 4d depicts the wind direction (vector) and magnitude (shaded) at 250 hPa. The subtropical high level jet is observed over Argentina near to a diffluence zone that covers Uruguay, southern Brazil, the northeast of Argentina and Paraguay, which may contribute to upward motion.

Conversely, the wind is significantly decelerating over this area, possibly leading to the opposite effect of the diffluence. This configuration may explain why wind divergence is not observed at 250 hPa in the region (not shown here). In addition, the trough observed at other levels also appears at 250 hPa slightly displaced to the northeast.

   The dynamical patterns presented in Figura 4 provide a favorable environment for convection development over the La Plata Basin. The trough, which extends vertically along the troposphere slightly tilted to northeast, moves toward the La Plata Basin,

contributing to low level convergence, and, therefore, upward motion. Furthermore, north wind at low levels advects heat and moisture, which provide an atmosphere favorable to MCS development.

   Synthetic satellite imagery is computed from model output in order to compare the simulations to GOES satellite observations. The synthetic brightness temperature is generated for the thermal channel 10.7 $\mu$m by using the Community Radiative Transfer Model (CRTM) that is described by Chen et al. (2008). Figure 5 shows these results at 06:00, 08:30 and 11:00 UTC

for different CCN concentrations, where the brightness temperature is shaded. The MCS area was delimited by contouring the brightness temperature equal to -32 °C (Maddox , 1980), in order to use this area as a mask for statistical analyses. By comparing the Fig. 5 against satellite observations (Fig. 3), we can see that the convective cells are slightly displaced toward the east, and the convection over the northeast of Argentina was not well simulated, and the lack of dense observational network in the region may partially explain the model's errors. However, the model presented results with reasonable accuracy

and was able to simulate the main system's life cycle, which allows for the study of aerosol impacts on the simulated MCS. Slight differences between the runs are noted, especially at 11:00 UTC. Experiments with more aerosols show smaller values of brightness temperature, which are located in different places for each simulation. These results suggest that CCNs can modify the microphysical structure of the system, although they do not seem capable of significantly changing its overall shape. Thus, a more detailed analysis is shown in the following paragraphs.

Figure 6 shows the precipitation results in terms of covered area, total accumulated and maximum rate as function of time. The covered area is computed by adding up the grid cell areas with precipitation for each 30 min; the total accumulated is basically the total of the precipitation occurred throughout time integration from 03:00 UTC; and the maximum rate is the maximum hourly accumulated precipitation in the finest grid domain for each 30 min. For these computations, we consider grid cells with at least 2 mm/h of precipitation. Precipitation shows different responses to CCN enhancements, depending on

the point of view. The smallest amounts of total accumulated precipitation as well as the precipitation covered area are seen in



the CN-Low run. As the CCN increases, larger amounts of total accumulated precipitation, covering bigger areas, are observed. On the other hand, the highest peak of the precipitation rate is seen at 09:00 UTC for CN-Med, followed by CN-Low, CN-High and CN-ExtHigh. The CN-Med experiment presents higher precipitation rates during most of the time.

Figure 7 shows the total integrated upwelling vapor flux at cloud base, taking into account only the grid cells with vertical velocity greater than zero. This variable is computed by multiplying the vertical velocity on the cloud base by the vapor mixing ratio right below and then integrating horizontally in area. We consider grid cells with cloud mixing ratio bigger than $1 \times 10^{-6}$ g/kg and positive vertical velocity. Slight enhancement of CCN concentrations is enough to significantly change the input of vapor into the system. Further CCN enhancement leads to stronger upwelling vapor flux but does not increase the flux proportionally, revealing non-monotonical behavior.

Updraft morphology is a key factor to understand the aerosol impact on cloud dynamics, as has been demonstrated in the literature (van den Heever et al., 2006; Carrió et al., 2011; Lebo and Morrison, 2014). Therefore, we computed the number and total area of updrafts for all runs as function of time (Fig. 8). To compute the updraft number we select the grid columns with mean vertical velocity (between 2 and 8 km height) greater than 0.9 m/s, hence, updrafts with a reasonable vertical extension are kept; second, we group neighboring columns; third, we keep grid cells within these columns that have vertical velocities greater than 1 m/s in order to find the updraft boundaries. The total updraft area was calculated by adding up the maximum horizontal area in the vertical of each localized updraft. Experiments with higher aerosol concentrations tend to produce more updrafts, which, in turn, cover larger areas. These differences become apparent for certain periods such as 04:00 – 05:00 (system formation), 07:00 – 08:00, 08:00 – 09:30 and 09:30 – 11:00 (mature stage). The greatest differences between CN-Low and the other experiments occur during the mature stage, where CN-Med, CN-High and CN-ExtHigh exhibit about 250 updrafts, whereas CN-low has about 200 updrafts.

When we compare Figs. 6, 7 and 8, similarities between these variables become evident and show an important link between cloud microphysics and dynamics. The number of updrafts and, consequently, their total covered area increase as aerosol concentration is enhanced. This effect results in higher values of total upwelling vapor flux at cloud base which feeds the condensation process. Hence, larger amounts of total accumulated precipitation are generated over a bigger area. It is important to note that the total accumulated covered area, the total upwelling vapor flux, the number of updraft cells and the total updraft area were computed using different approaches, which end up reinforcing the outcomes.

Volume integrals for liquid categories throughout time are computed for updraft greater than 5 m/s (Fig. 9). In order to avoid meaningless computation, the volume integrals for concentrations of each liquid hidrometeor class are weighted by their mass. The maximum quantities of liquid mass, resulting from warm processes, are noted between 08:30 and 09:00 UTC. As CCN increases, cloud droplets and supercooled droplets increase in mass and number, whereas drizzle and rain demonstrate an opposite pattern. This effect is widely discussed in the scientific literature. In fact, higher aerosol loadings nucleate larger number of cloud droplets, inducing a narrow cloud droplet spectrum, in other words, lots of droplets with smaller sizes. As a consequence, the cloud droplet collection and warm rain formation are suppressed, allowing cloud droplets to be thrust aloft to upper levels. Once cloud droplets reach the freezing levels, they immediately become supercooled and are eventually either collected by ice particles or freeze.



Figure 10 shows the mean vertical profiles of mixing ratio and number of particles for cloud, drizzle, rain and supercooled droplets, considering the grid columns whose vertical velocity is greater than 5 m/s (Fig. 10). The number concentration is weighted by the hydrometeor mass as in Fig. 9. By comparing Figs. 10 and 9, we can see that the aerosol effect observed in the volume integrals also clearly appears in the mean vertical profiles. In addition, we can see two well defined peaks in the

supercooled profile at 5 and 8 km, respectively. It is also possible to note two sharp decreases in the droplet mixing ratio; the first one, placed between 5 and 7 km, is probably related to the riming process; and the second one, above 8 km, is associated to homogeneous freezing since cloud droplets freeze instantaneously at this height (regardless of the size) where the temperature is below -35 °C.

Similar to Fig. 9, Fig. 11 depicts the volume integral of mass and number concentration for pristine and aggregates as a

function of time. The total pristine mass does not show a clear response to aerosol increase, although huge amounts of aerosol seem to contribute to a decrease in the pristine mass. On the other hand, pristine number concentration has a monotonic response to CCN enhancement. The total aggregate mass and number concentration present a nonlinear response to aerosol increase, as already mentioned. Therefore, a slight increase in CCN (CN-Med compared to CN-Low) favors aggregate formation. However, further enhancements have the same impact on CN-Med, in other words, total aggregate mass and concentration do not change

at the same rate as CCN particles increase. The average profiles for pristine and aggregates, given by Fig. 12 (constructed as Fig. 10), also captured the aerosol effect revealed in the volume integral. Moreover, above 8 km height, pristine mass and number concentration increase, while supercooled droplets decrease, indicating that ice particles are being formed by homogeneous freezing of cloud droplets.

Immersed within an environment rich in supercooled liquid water, aggregate particles are likely to grow by the rimming

process, rapidly allowing these ice particles to become graupel by collecting cloud droplets. Figure 13 (computed as Fig. 9) illustrates this effect clearly, with a monotonically increase in the graupel mixing ratio as CCN is enhanced. In contrast, hail decreases monotonically as aerosol increases because of the cloud diameter reduction in response to high CCN concentration, as explained earlier. Indeed, tiny cloud droplets are more likely to be deflected by the air flow around the hailstones since they do not have enough inertia. Figure 12, computed as Fig. 10, also shows this effect, being even clearer for the hail mixing ratio.

Figure 15 compares the vertical velocities produced by the experiments, where the left-hand panel shows an average of the three highest updraft peaks, whereas the right-hand panel shows the vertical velocity profile of the column with the strongest updraft. The updraft response to CCN number variation is highly nonlinear and depends on several factors. However, the updraft seems more intense for higher aerosol concentrations after 09:00 UTC (Fig. 15a), which might be associated to aggregate and graupel categories. The maximum vertical velocity is produced by the CN-ExtHigh experiment, followed by the CN-Med,

CN-Low and CN-High experiments (Fig. 15b). High aerosol loading may displace the updraft peak aloft as a consequence of homogeneous freezing enhancement (Carrió et al., 2014). This effect is due to riming suppression since cloud droplets are too small, being deflected by the air flow around the ice particles. Furthermore, riming suppression also causes a decrease in the updraft intensity below 10 km (Fig. 15b) owing to the lower latent heat realized.

Columns from the ground up to height of 2 km with downdraft greater than 1 m/s were averaged for buoyancy, rain mixing

ratio and downdraft speed (Fig. 16) for the time period between 9:00 and 10:00 UTC. The model outputs were saved every





5 min to compute these profiles. The CN-ExtHigh experiment stands out with the largest values of rain mixing ratio, leading to stronger buoyancy and downdraft than the CN-Low experiment. In addition, the other two experiments, CN-Med and CN-High, present similar behavior when compared to CN-Low, although less pronounced. The highest rain mixing ratio of the CN-ExtHigh experiment may suggest the contribution of ice melting. Indeed, higher amounts of rain favor stronger negative

buoyancy since the drops evaporate in an unsaturated environment, lowering the air temperature. This effect leads to heavier air parcels, which increase the vertical velocity toward the ground. Finally, stronger downdrafts intensify the low level convergence, which, in turn, may contribute to the formation and intensification of updrafts (Tao et al., 2007, 2012; Lebo and Morrison, 2014).

## 4   Conclusions

Many studies have shown the importance of studying the effect of biomass burning aerosols on cloud systems over the Amazon and neighboring regions (Andreae et al., 2004; Freitas et al., 2005; Koren et al., 2004, 2008; Martins et al., 2009; Martin et al., 2010; Artaxo et al., 2013; Gonçalves et al., 2015). Some authors suggest that these aerosols can be transported by low level flow to the La Plata Basin (Freitas et al., 2005). In this context, we performed four experiments with BRAMS-4.3, varying CCN concentration for an MCS case occurring over the La Plata Basin on 21 September 2010. The model was initialized with

idealized CCN profiles that were based on Freitas et al. (2005).

Generally, higher CCN loadings increase the concentration of cloud droplets, and, consequently, reduce the collision-coalescence efficiency. This effect allows more droplets to reach the freezing levels and become supercooled. The availability of supercooled cloud droplets is directly linked to the riming efficiency. The chances of these hydrometors being collected by ice particles increase since there are more supercooled droplets available. Therefore, larger aggregates and graupel mixing ra-

tios were observed as aerosol number concentration increases. Conversely, the hail mixing ratio decreased under larger aerosol loadings, which could be explained by the droplet size. Small cloud droplets are more likely to be deflected by the flow around hail particles, thereby reducing the riming efficiency. Then tiny supercooled droplets reach levels with temperatures below -35 °C, being instantaneously frozen.

The reduction of riming efficiency leads to a decrease in the updraft velocity below the homogeneous freezing level (below

10 km height) because less latent heat is released. However, since these small droplets were not collected by hail, they are capable of freezing homogeneously at upper levels. This effect increases the vertical velocity which, in turn, displaces the updraft peak to higher levels.

An invigoration of the downdrafts was verified for high CCN concentrations at low levels (below 2 km). The experiments with more CCN particles showed larger values of rain mixing ratio at low levels. Since there is more mass to evaporate, higher

values of negative buoyancy were noted. This phenomenon leads to an invigoration of the downdrafts. Stronger downdrafts helped to increase the number of updraft cells, and, consequently, the total area covered by them. With more updraft cells, higher amounts of total upwelling vapor flux were observed, favoring the condensation process. Associated with this effect, larger amounts of total accumulated precipitation over a bigger area were generated under higher concentration of CCN.





This paper is the first to study the impact of Amazon biomass burning aerosols on mesoescale convective system over the La Plata Basin from a numerical modeling point of view taking into account cloud microphysics – dynamics interactions. We showed an important link between aerosol microphysical effects, cloud dynamics and precipitation for the specific case of aerosols feeding an MCS in the La Plata Basin. Camponogara et al. (2014) observed a decrease in precipitation in the La Plata Basin associated with an increase in AOD (Aerosol Optical Depth) measured in the Amazon Basin as one of the possible scenarios detected in the data available for the period from 1999 up to 2012. They also detected a pattern where no aerosol effect on rainfall could be detected. Indeed, cloud-aerosol interactions are highly nonlinear, and different responses may occur, depending on environmental conditions as discussed in Sect. 1. Therefore, this issue needs further investigation, particularly with an extended number of cases, in order to fully understand the role of Amazon aerosols on mesoscale convective systems that take place over the La Plata Basin.

*Acknowledgements.* We thank the *Laboratório Nacional de Computação Científica* (LNCC) and the *Sistema Nacional de Processamento de Alto Desempenho* (SINAPAD) as well as the *Instituto Nacional de Pesquisas Espaciais* (INPE) and the *Centro de Previsão de Tempo e Estudos Climáticos* (CPTEC) for making available their computational resources to perform numerical simulations. We also thank Dr. Saulo Freitas for his support with respect to BRAMS. This research was supported by São Paulo Research Foundation (FAPESP), grant no. 2009/15235-8, 2012/08115-9 and 2014/05351-9.





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





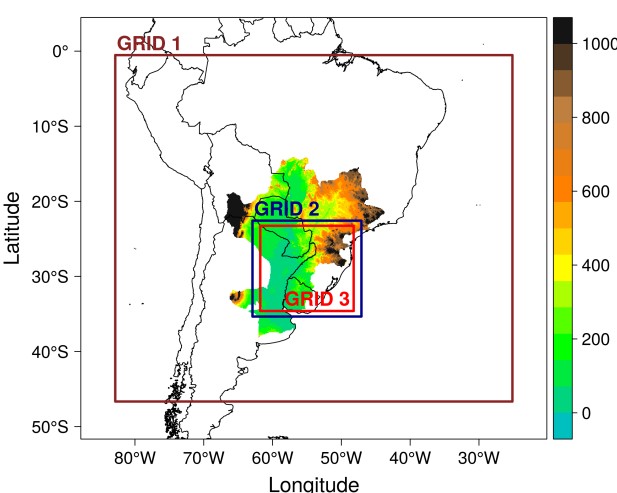

**Figure 1.** Model domain for 16 km (brown box), 4 km (blue box) and 2.5 km (red box) of grid spacing. The topography elevation of the La Plata Basin is shaded.



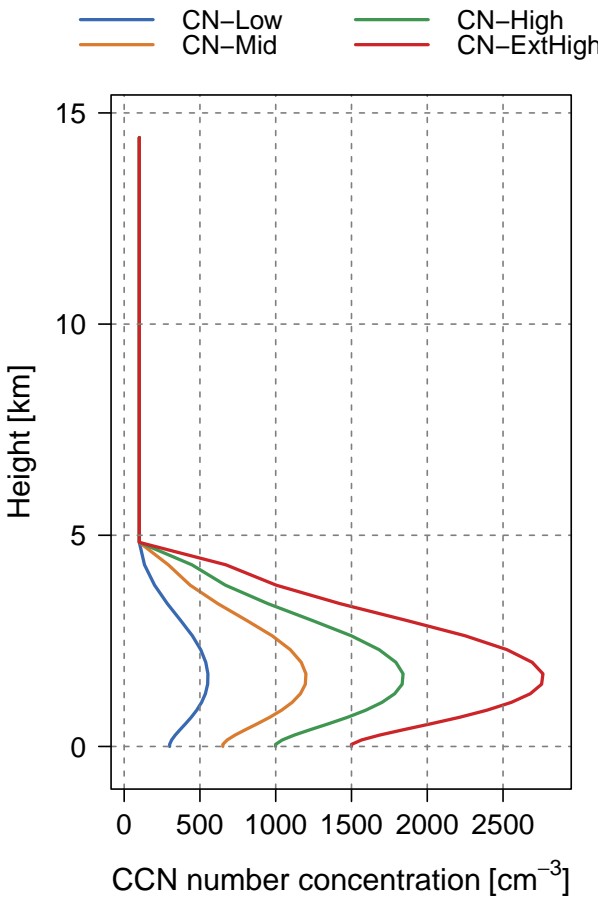

**Figure 2.** Vertical profiles of CCN number concentrations used to initialize the model.

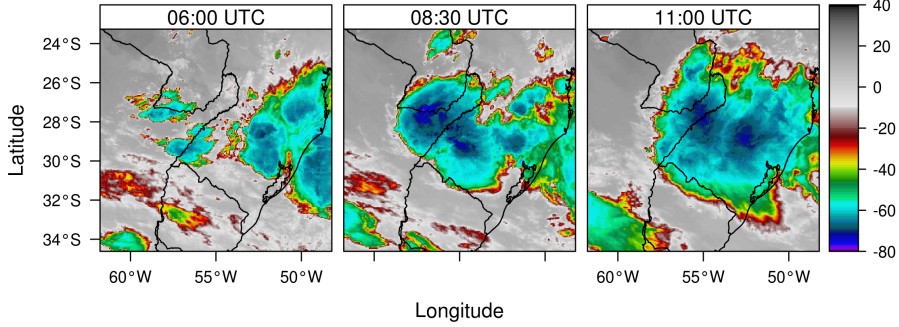

**Figure 3.** Enhanced infrared satellite images from GOES 12 for an MCS case observed over the La Plata Basin on 21 September 2010. Colors indicate infrared temperatures.





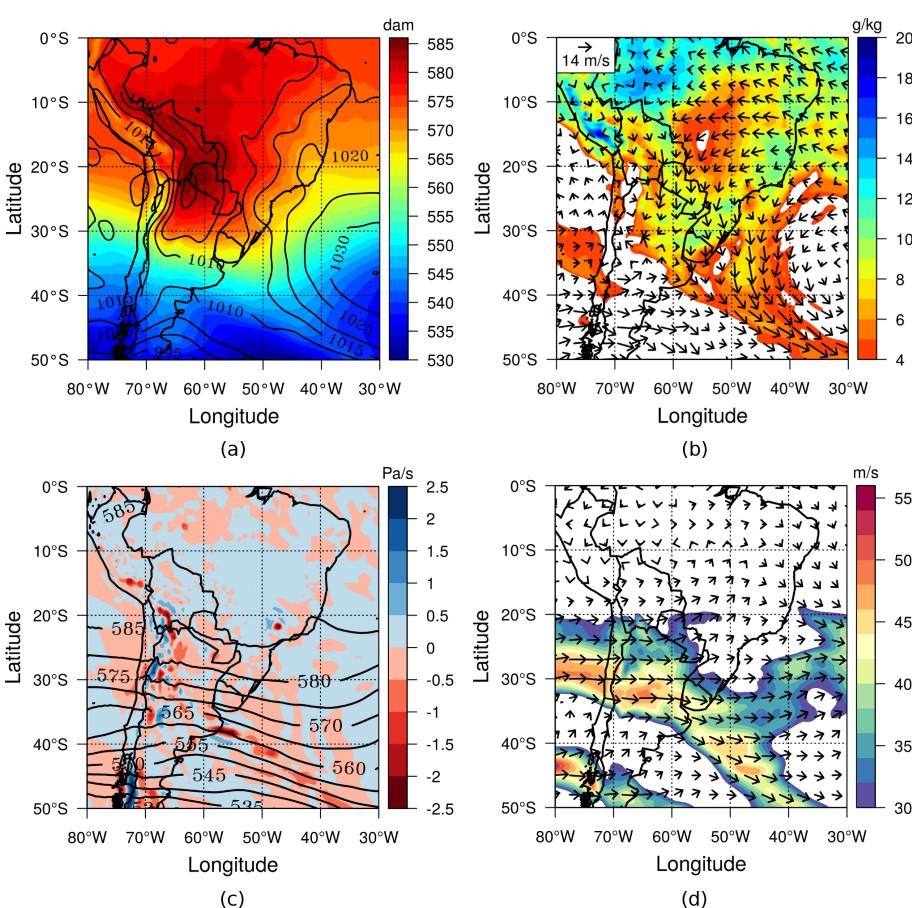

**Figure 4.** Synoptic fields on 21 September 2010 at 00:00 UTC: (a) sea level pressure (contour lines) and thickness (shaded); (b) wind (vectors) and specific humidity (shaded) at 850 hPa; (c) geopotential height (contour) and vertical p–velocity (shaded) at 500 hPa; and (d) wind direction (vector) and magnitude (shaded) at 250 hPa.



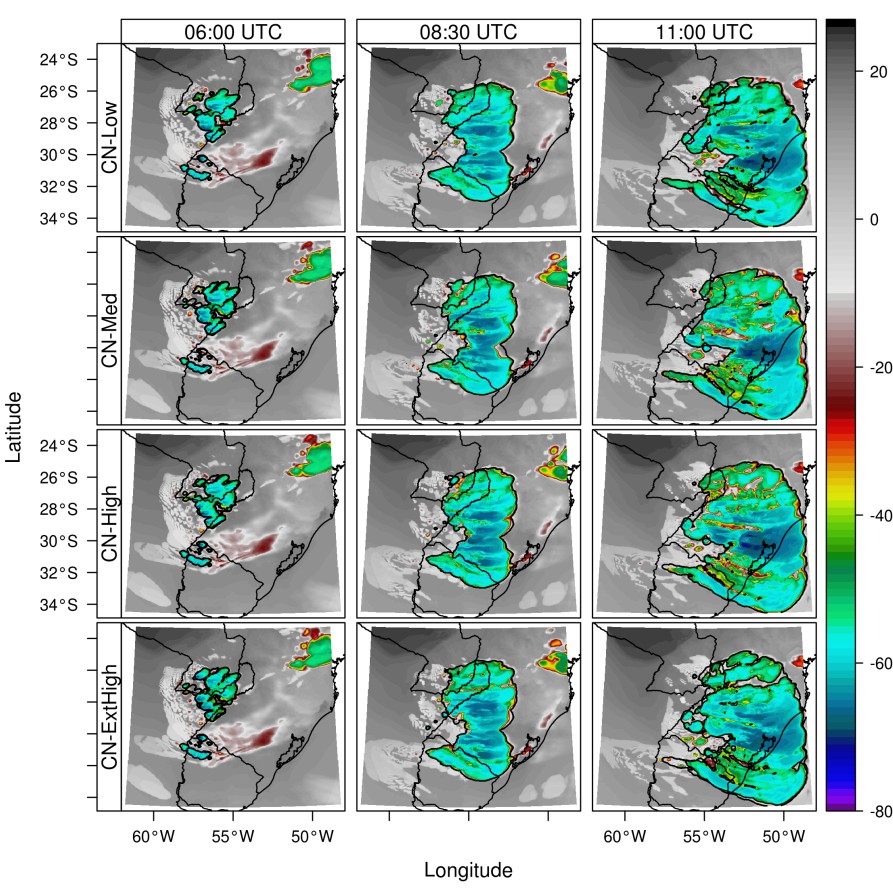

**Figure 5.** Synthetic infrared satellite images for the CN-Low, CN-Med, CN-High and CN-ExtHigh experiments at 06:00 (left panels), 08:30 (middle panels) and 11:00 UTC (right panels). Colors indicate brightness temperature at channel $10.7\,\mu m$. The black contour line refers to the brightness temperature equal to -32 °C, which delimits the MCS area (Maddox , 1980).





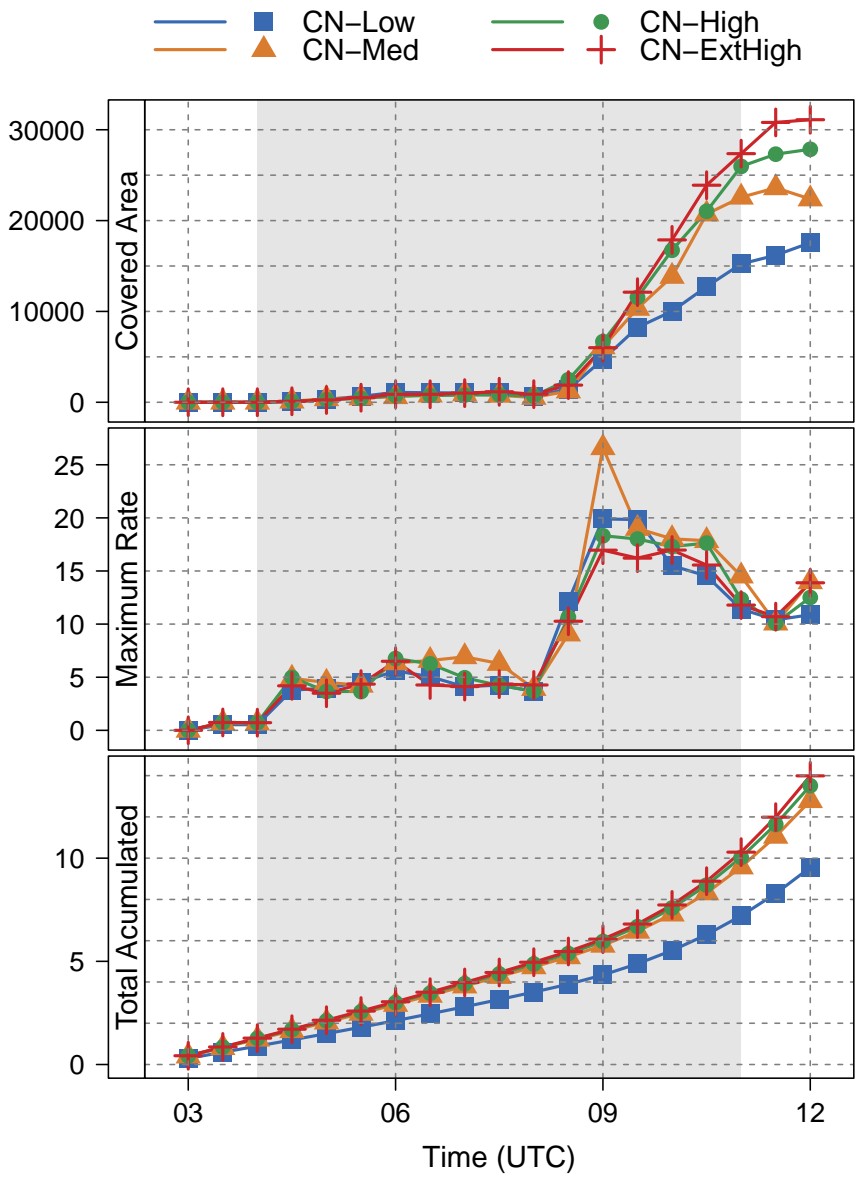

**Figure 6.** Precipitation covered area (km$^2$), maximum precipitation rate (mm/h) and total accumulated precipitation ($1 \times 10^5$ mm) as a function of time for the CN-Low (blue), CN-Med (marigold), CN-High (green) and CN-ExtHigh (red) experiments. The shaded area in grey represents the time period that the entire MCS is within the grid domain.




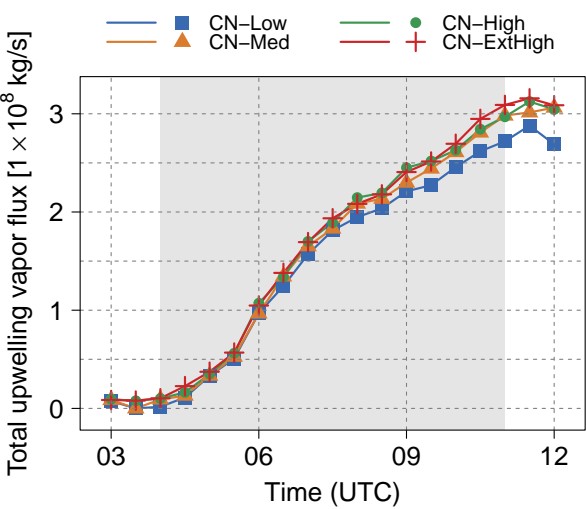

**Figure 7.** Total upwelling vapor flux at cloud base as function of time for the CN-Low (blue), CN-Med (marigold), CN-High (green) and CN-ExtHigh (red) experiments. The shaded area in grey represents the time period that the entire MCS is within the grid domain.





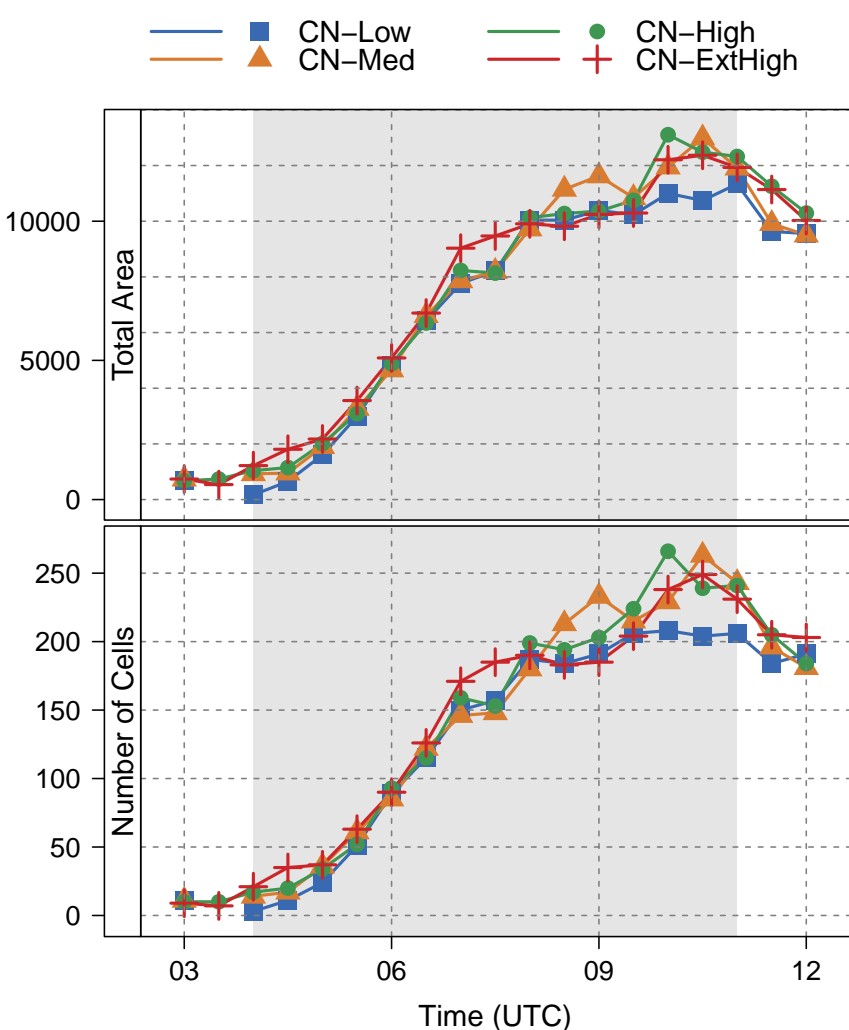

**Figure 8.** Total area of updraft (km$^2$) and number of updraft cells as a function of time for the CN-Low (blue), CN-Med (marigold), CN-High (green) and CN-ExtHigh (red) experiments. The volume integral of concentrations is weighted by their respective mass. The shaded area in grey represents the time period that the entire MCS is within the grid domain.



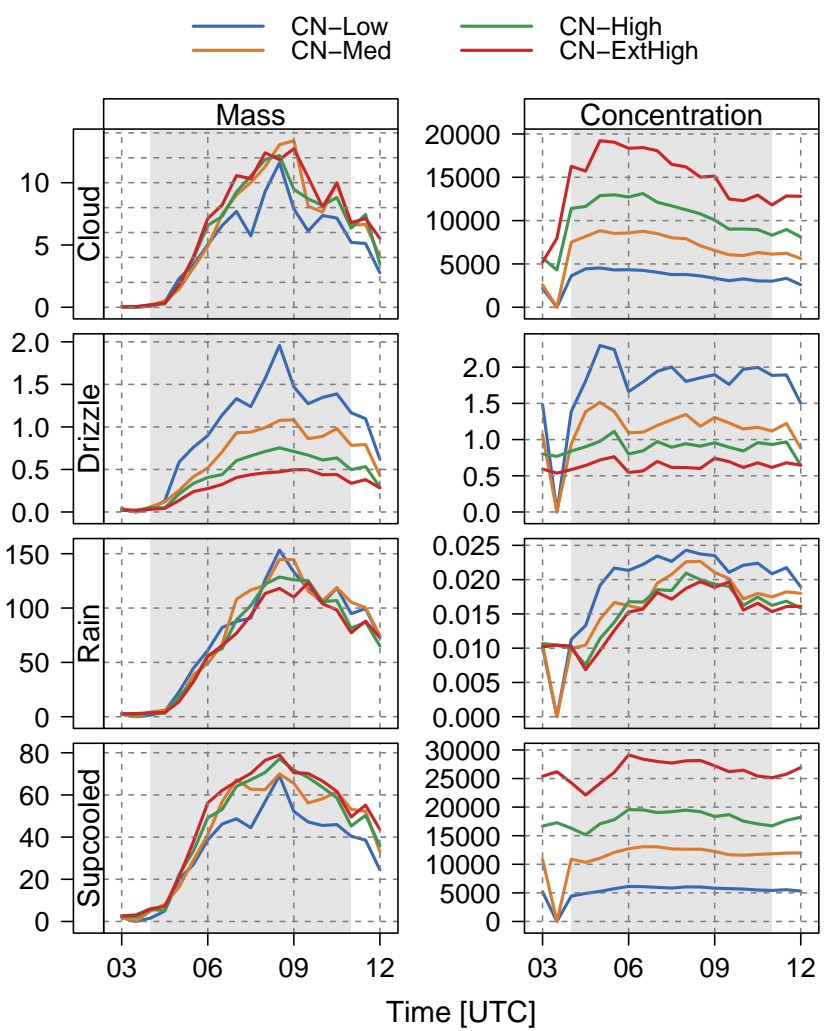

**Figure 9.** Volume integrals of cloud, drizzle, rain and supercooled cloud in terms of mass ($1 \times 10^8$ kg) and number concentration ($1 \times 10^5$ m$^{-3}$) throughout time for the CN-Low (blue), CN-Med (marigold), CN-High (green) and CN-ExtHigh (red) experiments. Only grid columns with updraft greater than 5 m/s are considered in the computation. The number concentrations are weighted by their respective mass. The shaded area in grey represents the time period that the entire MCS is within the grid domain.





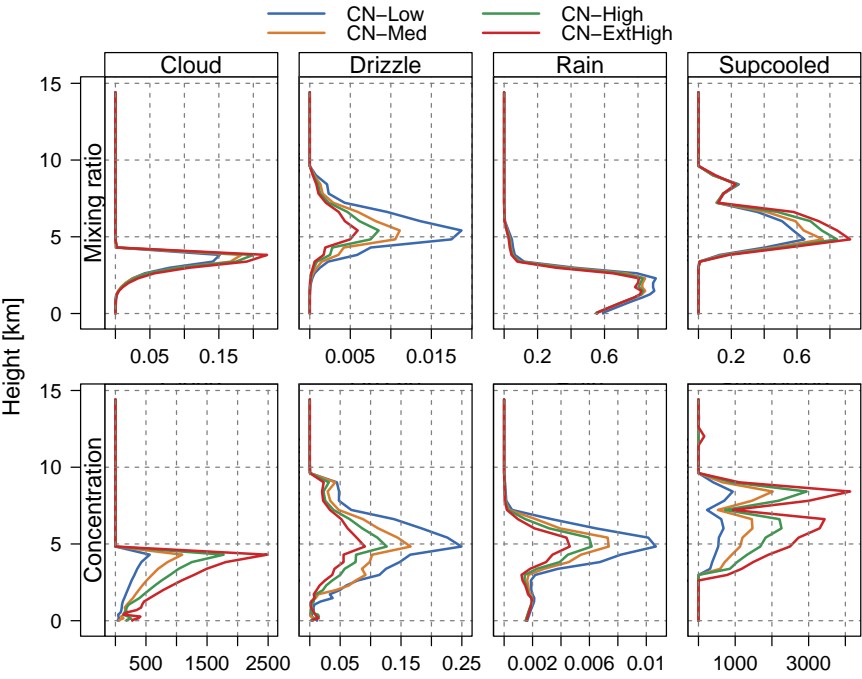

**Figure 10.** Mean vertical profiles of mixing ratio (g/kg) and number concentration ($1 \times 10^6$ m$^{-3}$) for cloud, drizzle, rain and supercooled cloud for the CN-Low (blue), CN-Med (marigold), CN-High (green) and CN-ExtHigh (red) experiments. Only grid columns with updraft greater than 5 m/s are considered in the computation.





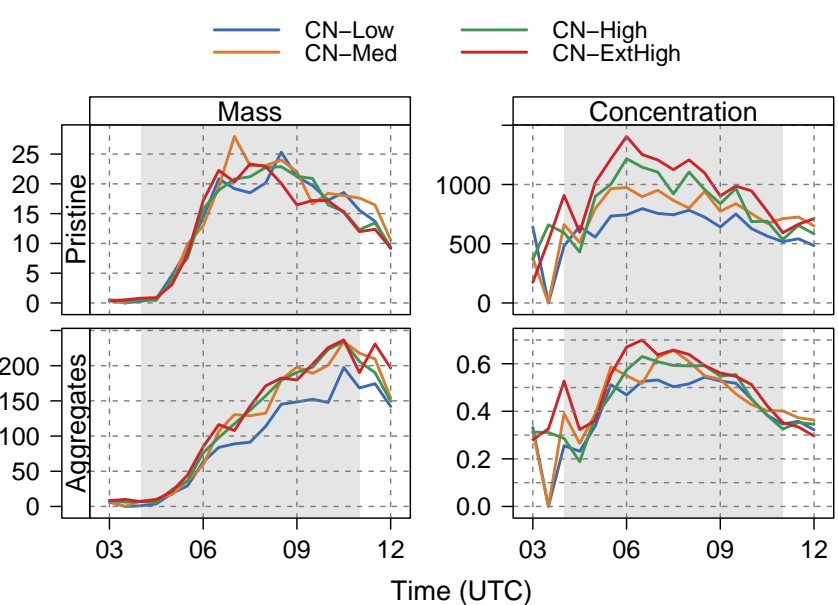

**Figure 11.** As in Figure 9, but for pristine and aggregate categories.





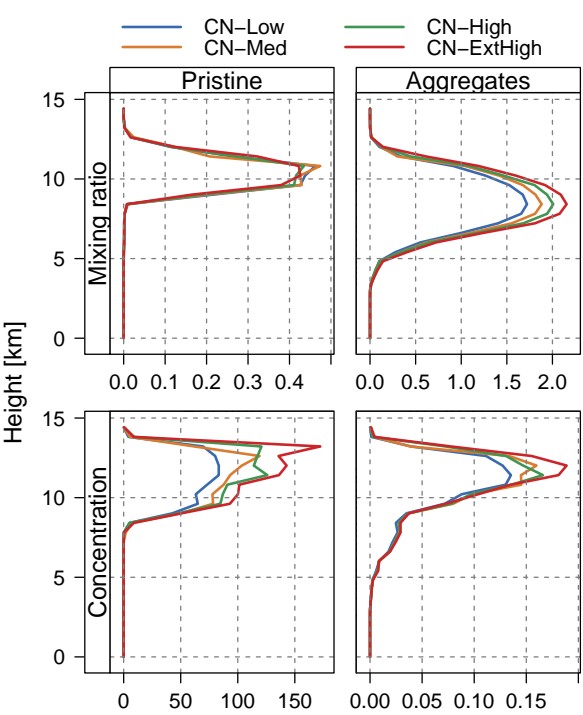

**Figure 12.** As in Figure 10, but for pristine and aggregate categories.



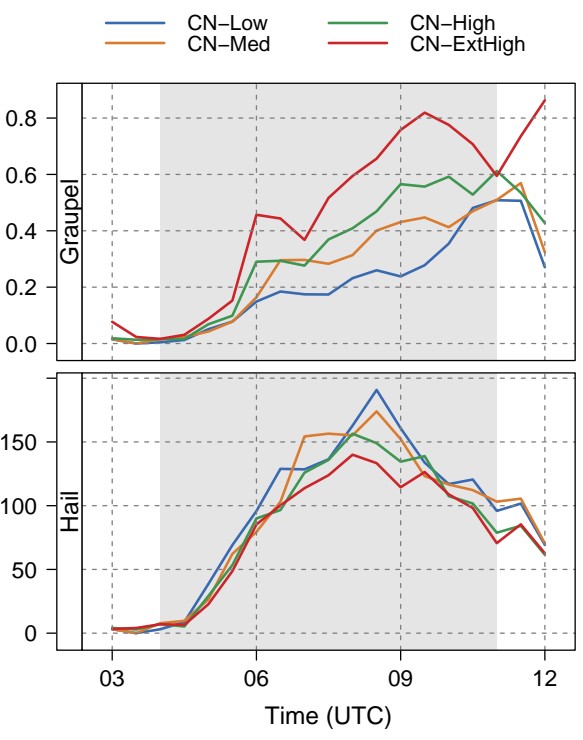

**Figure 13.** As in Figure 9, but for graupel and hail mixing ratios.





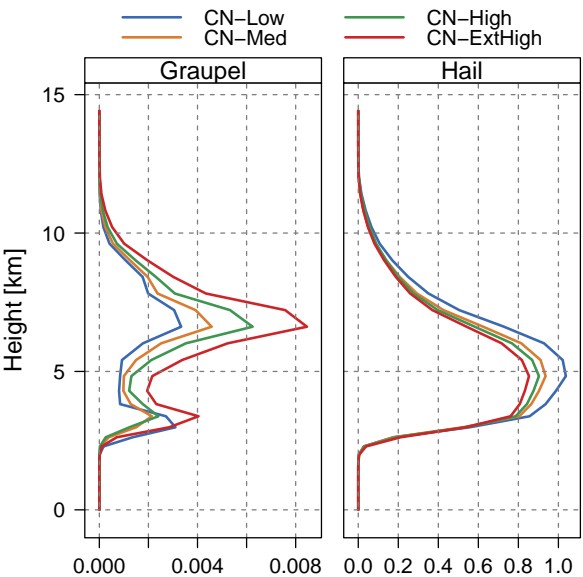

**Figure 14.** As in Figure 10, but for graupel and hail mixing ratios.

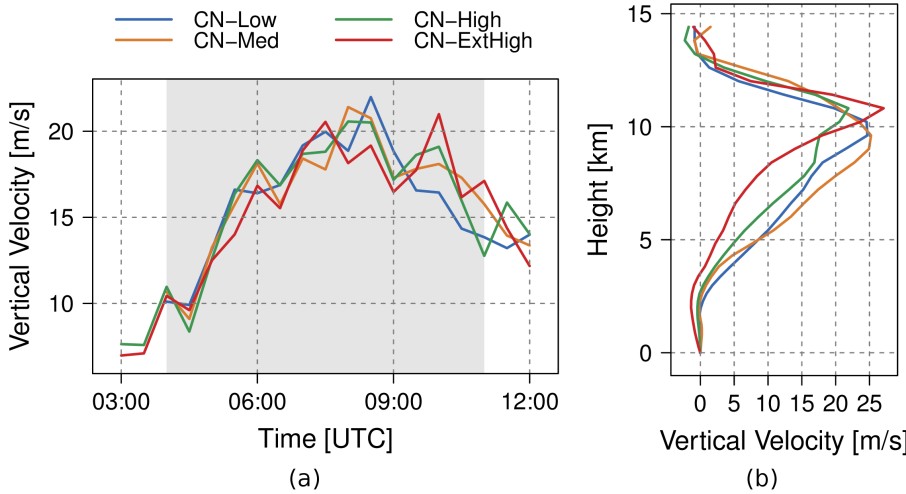

**Figure 15.** Average of the three highest updraft peaks throughout time (a) and strongest updraft profile (b) for the CN-Low (blue), CN-Med (marigold), CN-High (green) and CN-ExtHigh (red) experiments. The shaded area in grey represents the time period that the entire MCS is within the grid domain.





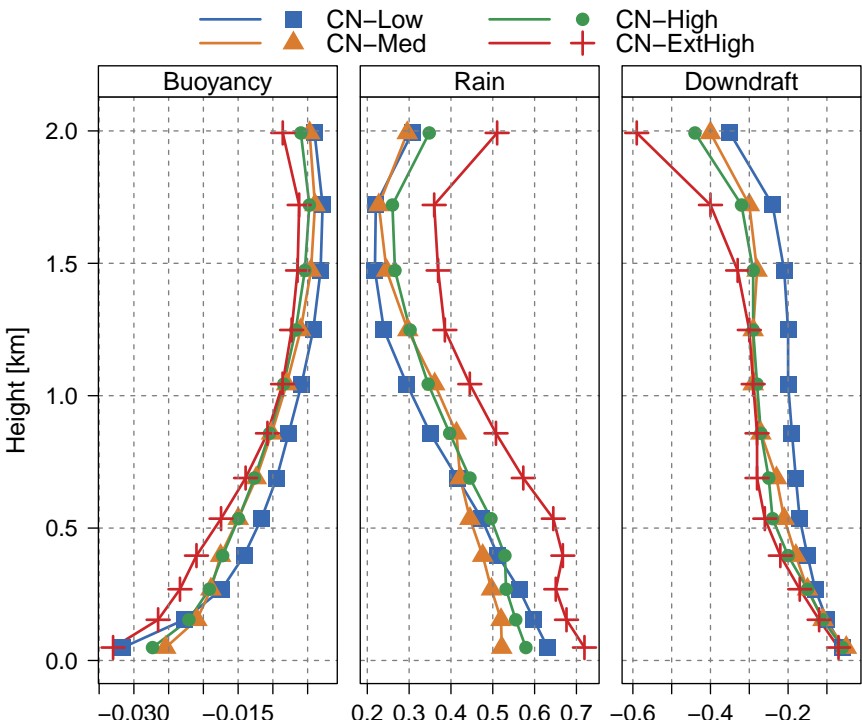

**Figure 16.** Averaged buoyancy, rain mixing ratio and downdraft between 9:00 and 10:00 UTC for the CN-Low (blue), CN-Med (marigold),
CN-High (green) and CN-ExtHigh (red) experiments. The model outputs were saved every 5 min to compute these profiles. Only vertical
velocities smaller than -1 m/s are considered for the average.




**Table 1.** BRAMS main configuration.

| | |
|---|---|
| Number of points for lateral boundary nudging | 5 |
| Nudging time scale for lateral boundary | 1800 s |
| Nudging time scale for the top of domain | 10800 s |
| Lateral boundary condition | Klemp and Wilhelmson (1978) |
| Shortwave/Longwave radiation parametrization | Chen and Cotton (1987) |
| Turbulence parametrization | Mellor and Yamada (1982) |
| Convective parametrization (activated only for grid 1) | Grell (1993) |