# Peer review of "Biomass burning CCNs enhance the dynamics of a Mesoscale Convective System over the La Plata Basin: a numerical approach"

_Atmospheric Chemistry and Physics, 2017_

## Referee Comment (RC1) · Anonymous Referee #1 · 1 Jun 2017

This study simulates the effect of biomass burning aerosols as cloud condensation nuclei on a deep convective cloud system over the Amazon using a cloud-resolving model with a two-moment bulk cloud microphysics. The finding about the aerosol invigoration effect essentially agrees with previous modeling studies (Khain et al., 2005; Li et al., 2009; Wang et al., 2011; Fan et al., 2016) with the similar topics. Considering there are abundant observations emerging from recent field campaigns over Amazon, a modeling study like this one would be of interest to the community. The paper can be accepted by ACP after authors can prove the robustness of the simulated aerosol effect and better illustrate some critical microphysical processes following the CCN perturbations.

[Figure]

1. The numerical experiments are based on one set of initial and lateral boundary conditions from NCEP Reanalysis. Hence, readers can easily raise a question like how sensitive the simulated aerosol effect is to the meteorological fields? One approach to address such concern is to carry out additional ensemble simulations by reasonably perturbing initial and boundary meteorological fields. With those analyses, the authors can obtain spread of simulated aerosol signals and justify the robustness of the aerosol effect. The similar approach has been used in the previous studies (Wang et al., 2014, Lin et al., 2016)

2. Page 7 lines 5-9, the changes in upwelling vapor flux at cloud base are not well explained. Since the aerosol induced additional latent heat release is expected to occur well below the cloud base, the changes at the cloud base level may indicate the modification on the whole circulations in the cloud such as low level convergence.

3. Fig. 10 and 12, to better reveal the heterogeneous versus homogeneous ice nucleation, I suggest to provide 0 degree and -35 degree isothermal lines.

4. Page 8 lines 20-24, more discussions are still needed to explain why CCN leads to an increase in graupel but a decrease in hail. Both hydrometeors are formed mainly by the riming processes. The authors only talk about cloud droplet size, but neglect the influence of the raindrop size. The finding about distinctive responses of graupel and hail is important, as some microphysics do not consider hail explicitly. A plot/table to show the riming efficiency is desired.

5. Since the latent heat release plays a central role in the aerosol invigoration effect, it is extremely valuable to display the spatial distribution of latent heat explicitly.

6. References mentioned above and suggested to be discussed in the manuscript:

Fan, J., et al. "Review of Aerosol-Cloud Interactions: Mechanisms, Significance, and Challenges", J. Atmo. Sci. 73 (11), 4221-4252 (2016)

Khain, A., et al. "Aerosol impact on the dynamics and microphysics of deep convective

clouds", Q. J. Royal Meteo. Soc. 131(611), 2639-2663 (2005)

Li, G., et al. "The effects of aerosols on development and precipitation of a mesoscale squall line", J. Geophys. Res. Atmos., 114, D17205, (2009)

Lin, Y., et al. "Distinct Impacts of Aerosols on an Evolving Continental Cloud Complex during the RACORO Field Campaign", J. Atmo. Sci. 73(9), 3681-3700 (2016)

Wang, Y., et al. "Long-term impacts of aerosols on precipitation and lightning over the Pearl River Delta megacity area in China", Atmo. Chem. Phys., 11(23), 12421-12436, (2011)

Wang, Y., et al. "Distinct Effects of Anthropogenic Aerosols on Tropical Cyclones", Nature. Clim. Change, 4, 368–373 (2014)

---

## Referee Comment (RC2) · Anonymous Referee #2 · 11 Jul 2017

This article describes a modeling study of aerosol impacts on an MCS, using a case study in the La Plata Basin of South America. This is an interesting region to study aerosol impacts, as very strong convection tends to occur here, and it can also be greatly affected by the burning season in the Amazon. Convection in South America has been studied more and more lately, and this work has the potential to add some interesting considerations with regard to the effect of aerosols on storm dynamics. However, I felt the results presented were fairly superficial and did not examine any dynamics specific to the structure or organization of an MCS. I would consider accepting this paper for publication if some effort were put into further analysis.

[Figure]

1) The authors spend some time describing the features of an MCS in the introductory text, and then touch on none of them during the analysis and discussion. For example: there is a description of previous work on cold pool vs. shear dynamics, yet aside from a quick plot of buoyancy, the authors didn't examine how the actual storm organization changed with increased CCN. Were the cold pools deeper, colder, more widespread? Did this affect the longevity of individual convective cores or help to promote the growth of new ones? Could any difference be seen in the shape/tilt of the updrafts with a different shear/cold pool balance?

2) Most of the results were presented as domain averages, which leaves out a lot of details. The particularly interesting thing about an MCS is its complex structure, and averaging over all of this may gloss over important features. It would be quite useful, for instance, to know if any notable difference occurs in the storm anvil vs the convective core, or between convective and stratiform precipitation.

3) The calculation of updrafts used in Figs 8,15 are not explained well. Are you considering an 'updraft' as a core consisting of multiple model columns? It is not clear how Fig 8 A and B are different. More analysis of the behavior of updraft cores would be beneficial, but the authors need to be clear in their definitions.

A few suggested references:

Clavner, M., L. D. Grasso, W. R. Cotton, and S. C. van den Heever, 2017: The response of simulated mesoscale convective system to increased aerosol pollution: Part II: Derecho characteristics and intensity in response to increase pollution, Atmos. Res., In Press, http://dx.doi.org/10.1016/j.atmosres.2017.06.002

Fan, J., D. Rosenfeld, Y. Ding, L. R. Leung, and Z. Li (2012), Potential aerosol indirect effects on atmospheric circulation and radiative forcing through deep convection, Geophys. Res. Lett., 39, L09806, doi:10.1029/2012GL051851.

Saleeby, S. M., S. C. van den Heever, P. J. Marinescu, S. M. Kreidenweis, and P. J.

[Figure]

DeMott (2016), Aerosol effects on the anvil characteristics of mesoscale convective systems, J. Geophys. Res. Atmos., 121, 10,880–10,901, doi:10.1002/2016JD025082.

Seigel, R.B. and S.C. van den Heever, 2013: Squall-Line Intensification via Hydrometeor Recirculation. J. Atmos. Sci., 70, 2012–2031, https://doi.org/10.1175/JAS-D-12-0266.1

---

## Author Comment (AC1) · 29 Sep 2017

We thank the referee for the criticism and suggestions that have help us improve the paper.

1. The numerical experiments are based on one set of initial and lateral boundary conditions from NCEP Reanalysis. Hence, readers can easily raise a question like how sensitive the simulated aerosol effect is to the meteorological fields? One approach to address such concern is to carry out additional ensemble simulations by reasonably perturbing initial and boundary meteorological fields. With those analyses, the authors can obtain spread of simulated aerosol signals and justify the robustness of the aerosol

effect. The similar approach has been used in the previous studies (Wang et al., 2014, Lin et al., 2016).

Thank you for the suggestion. This is an interesting idea we look forward to following this line of work in the next study. We also included this idea as future work in the paper.

2. Page 7 lines 5-9, the changes in upwelling vapor flux at cloud base are not well explained. Since the aerosol induced additional latent heat release is expected to occur well below the cloud base, the changes at the cloud base level may indicate the modification on the whole circulations in the cloud such as low level convergence.

We have improved the explanation about the upwelling vapor flux. We appreciated the suggestion.

3. Fig. 10 and 12, to better reveal the heterogeneous versus homogeneous ice nucleation, I suggest to provide 0 degree and -35 degree isothermal lines.

We added a shaded area that corresponds to the layer between 0 and -35 degree isothermal lines. Thank you for the idea.

4. Page 8 lines 20-24, more discussions are still needed to explain why CCN leads to an increase in graupel but a decrease in hail. Both hydrometeors are formed mainly by the riming processes. The authors only talk about cloud droplet size, but neglect the influence of the raindrop size. The finding about distinctive responses of graupel and hail is important, as some microphysics do not consider hail explicitly. A plot/table to show the riming efficiency is desired.

We mainly focused on the effect of CCN on cloud droplet spectra as the collision efficiencies between liquid and ice particles dramatically vary when changing the drop size within the cloud droplet range (i.e., several orders of magnitude below 40 microns in radius), compared to those coming from potential changes in raindrops. For small ice particles interacting with supercooled liquid particles the variation of collision effiInteractive comment

ciencies linked to changes in droplet size are expected to be far more important that the collision efficiencies impact within the raindrop size changes. Actually, the circulation around the falling raindrop can invert the increasing response making collision efficiencies slightly lower when increasing raindrop size. In the case of graupel/hail collecting liquid particles, changes in collision efficiencies produced by raindrop range size changes are expected to be almost negligible. Moreover, in this stage of growth, thermodynamic limits are expected to play a dominant role compared to the buck amount of liquid per unit time impacting the hail/graupel particle. We have included in the atmospheric model section (line 30-31) that the riming efficiencies are described in detail by Saleeby et al. (2008).

5. Since the latent heat release plays a central role in the aerosol invigoration effect, it is extremely valuable to display the spatial distribution of latent heat explicitly.

Thank you for the suggestion. We have added the total precipitation over the space and a figure with the ratio between convective/stratiform precipitation in order to give more detail about MCS structure.

6. References mentioned above and suggested to be discussed in the manuscript:

Thank you for suggesting the references, we have included them in the text.

ACPD

---

## Author Comment (AC2) · 29 Sep 2017

We thank the referee for the criticism and suggestions that have help us improve the paper.

*1. The authors spend some time describing the features of an MCS in the introductory text, and then touch on none of them during the analysis and discussion. For example: there is a description of previous work on cold pool vs. shear dynamics, yet aside from a quick plot of buoyancy, the authors didn't examine how the actual storm organization changed with increased CCN. Were the cold pools deeper, colder, more widespread? Did this affect the longevity of individual convective cores or help to promote the growth*

*of new ones? Could any difference be seen in the shape/tilt of the updrafts with a different shear/cold pool balance?*

Dear referee, we have isolated the cold pools in order to examine the aerosol impacts on them. Unfortunately we have had several problems in isolating them due to the complexity of the simulation:

- The model grid spacing is 2.5 km. We believe that it would be easier to isolate the cold pools in finer grids;
- The topography of the domain is complex thus upslope and downslope circulation combine with cold pools and confounding the distinction between the two.
- The horizontal temperature gradient was quite strong due to a cold front approach, thus making it hard to distinguish the effect of the cold pool.

The automatic cold pool localization did not converge due to these reasons. This will be an issue for future research.

*2. Most of the results were presented as domain averages, which leaves out a lot of details. The particularly interesting thing about an MCS is its complex structure, and averaging over all of this may gloss over important features. It would be quite useful, for instance, to know if any notable difference occurs in the storm anvil vs the convective core, or between convective and stratiform precipitation.*

Thank you for the suggestion. We have computed and included in the paper the ratio between convective and stratiform precipitation in order to show with more detail features about MCS's precipitation.

*3) The calculation of updrafts used in Figs 8,15 are not explained well. Are you considering an 'updraft' as a core consisting of multiple model columns? It is not clear how Fig 8 A and B are different. More analysis of the behavior of updraft cores would be beneficial, but the authors need to be clear in their definitions.*

We appreciated your suggestions. We have improved the text in order to make it clear.

[Figure]

The changes are highlighted.

---

## Editor Decision (ED1)

In their response to reviewers comments, the authors did not adequately address all of the points. Please provide further responses to the following comments:

**Referee #1, comment 1.** The referee suggested the authors carry out additional ensembles to evaluate the robustness of the simulated aerosol effects. This seems like a reasonable request. Is there some reason the authors decided not to do this?

**Referee #1, comment 4.** The referee requested additional explanation of why increasing CCN leads to an increase in graupel but a decrease in hail. The authors responded with a discussion of collision efficiency dependence on droplet size. It is not clear to me that they answered the referee's question. Moreover, I did not find any additional explanation in the revised manuscript. If the authors feel there is no need for additional explanation, they should explain why.

**Referee #1, comment 5.** The referee asked for a display of latent heat. The authors responded by stating they added plots of precipitation. This isn't what the referee asked for. If the authors think plots of latent heat are not appropriate, they should explain why.

**Referee #2, comment 1.** The referee asks for several details about changes in the features of the MCS with increasing CCN, including cold pool structure, longevity of individual cores, and shape/tilt of updrafts. The authors respond by stating that their analysis of cold pools failed, but they don't address the other issues. A complete response should be provided.

---

## Author Response (AR2)

**Response to Co-Editor**

We thank the Co-Editor for the criticism and suggestions that have contributed to improve the paper.

*Referee #1, comment 1. The referee suggested the authors carry out additional ensembles to evaluate the robustness of the simulated aerosol effects. This seems like a reasonable request. Is there some reason the authors decided not to do this?*

It is not feasible to do it for this paper because it is a major change that manly requires additional funding for an expiring project since we need a computer user and a lot of time. However, we considered the reviewer suggestion for future work as it stands in the paper conclusions.

*Referee #1, comment 4. The referee requested additional explanation of why increasing CCN leads to an increase in graupel but a decrease in hail. The authors responded with a discussion of collision efficiency dependence on droplet size. It is not clear to me that they answered the referee's question. Moreover, I did not find any additional explanation in the revised manuscript. If the authors feel there is no need for additional explanation, they should explain why.*

The response in hail to CCN is very complex, nonlinear (nonmonotonically) even in a single cloud systems and, as and additional explanation, we included in the manuscript the following sentences (l. 4-7, p. 10; highlighted in blue in the paper):

The interaction with hail are quite indirect and highly nonlinear (Carrio et. al, 2014). This indirect response is linked to the collision efficiencies of super cooled liquid droplets that can reverse the response of hail to CCN. Modest enhancements of CCN lead to an increased availability of super cooled liquid water favoring the growth of hail; further enhancement of CCN leads to very small supercooled droplets whose collection efficiencies (with hail) dramatically decrease.

*Referee #1, comment 5. The referee asked for a display of latent heat. The authors responded by stating they added plots of precipitation. This isn't what the referee asked for. If the authors think plots of latent heat are not appropriate, they should explain why.*

As discussed by Cotton and Anthes (1989), the precipitation over a surface is linked to the vertical integral of $Q_2$ , the apparent moisture sink,which indicates changes of the air temperature due to moisture sink (we have added this explanation in the paper highlighted in blue at p. 7 and l. 31-34). Regions with more precipitation are expected to have higher values of vertical integrated $Q_2$. Therefore, as the referee asked for a horizontal distribution of latent heat release, we thought that would be appropriated to show it in terms of total precipitation. Moreover, the latent heat releases play an important role in the cloud-CCN interactions, relevant in this study.

*Referee #2, comment 1. The referee asks for several details about changes in the features of the MCS with increasing CCN, including cold pool structure, longevity of individual cores, and shape/tilt of updrafts. The authors respond by stating that their analysis of cold pools failed, but they don't address the other issues. A complete response should be provided.*

It could be easily done for highly organized or single cell system in idealized experiments. In our case, it was not possible to isolate cold pools because of the horizontal heterogeneity of our large domain and the fact we studied a complex cloud system. However, we performed an analysis on the downdrafts that generates cold pools and we isolated the updraft cells, computing their number and size. The results suggest a link between downdrafts intensity and updraft cell quantity; that is, as CCN number increases, more intense are the downdrafts and more updraft cells are observed. We have highlighted this link at p. 10 and l. 26-29. Moreover, we did not address the other question of the comment 1 because all the questions were related to cold pools.

**References**

[revised manuscript text omitted]